# Nature-Based Citizen Science as a Mechanism to Improve Human Health in Urban Areas

**DOI:** 10.3390/ijerph19010068

**Published:** 2021-12-22

**Authors:** Craig R. Williams, Sophie M. Burnell, Michelle Rogers, Emily J. Flies, Katherine L. Baldock

**Affiliations:** 1UniSA Clinical and Health Science, University of South Australia, GPO Box 2471, Adelaide, SA 5001, Australia; a1846305@student.adelaide.edu.au (S.M.B.); Michelle.Rogers@unisa.edu.au (M.R.); 2School of Natural Sciences, University of Tasmania, Private Bag 55, Hobart, TAS 7001, Australia; emily.flies@utas.edu.au; 3UniSA Allied Health and Human Performance, University of South Australia, GPO Box 2471, Adelaide, SA 5001, Australia; Katherine.Baldock@unisa.edu.au

**Keywords:** natural environments, urbanisation, public health, policy, Citizen Science

## Abstract

The world is becoming increasingly urbanised, impacting human interactions with natural environments (NEs). NEs take a number of forms, ranging from pristine, modified, to built NEs, which are common in many urban areas. NEs may include nature-based solutions, such as introducing nature elements and biological processes into cities that are used to solve problems created by urbanisation. Whilst urbanisation has negative impacts on human health, impacting mental and physical wellbeing through a number of mechanisms, exposure to NEs may improve human health and wellbeing. Here, we review the mechanisms by which health can be improved by exposure to NEs, as explained by Stress Reduction Theory, Attention Restoration Theory, and the ‘Old Friends’/biodiversity hypothesis. Such exposures may have physiological and immunological benefits, mediated through endocrine pathways and altered microbiota. Citizen Science, which often causes exposure to NEs and social activity, is being increasingly used to not only collect scientific data but also to engage individuals and communities. Despite being a named component of scientific and environmental strategies of governments, to our knowledge, the intrinsic health benefits of Citizen Science in NEs do not form part of public health policy. We contend that Citizen Science programs that facilitate exposure to NEs in urban areas may represent an important public health policy advance.

## 1. Introduction

Human habitats have become increasingly urbanised, and these urban environments are strongly divergent from the habitats in which humans have spent most of their evolutionary history. Subsequently, human health has been affected by urbanisation including increased prevalence of allergic, autoimmune, inflammatory, metabolic and infectious ‘urban-associated diseases’ [1]. However, compared to remote and regional communities, income and access to health services are often higher for people living in cities, as are some health indicators such as longevity and total disease burden [2] demonstrating a complex relationship between urbanisation and human health. Within cities, living near green spaces and natural environments (NEs) typically confers health benefits ranging from better mental health and wellbeing to reduced overall mortality [3]. However, adding to this urban health complexity are findings from some studies showing increased mortality in cities with greater overall ‘greenness’ [4], and tensions between green space creation, urban gentrification and social inequity [5]. 

Categorising an environment as natural may be subjective, as it may have both natural and unnatural qualities. Indeed, ‘nature’ and ‘natural’ are conceptual terms that may be understood in different ways [6]. In reality, even environments perceived as natural may have some level of disturbance or modification, and thus exist on a spectrum between wild and pristine (‘organic’ NEs), through to partially natural (e.g., urban forest preserve with trails) or heavily disturbed (e.g., managed, grassy parkland). NEs may also be re-created through the establishment of community gardens and green roofs (‘built’ NEs). For this paper, we understand natural environments (NEs) to be any vegetated area with less built aspects and anthropogenic disturbance compared to surrounding areas; this definition includes both ‘organic’ and ‘built’ forms of green open spaces, gardens, parks, reserves and other forms of ‘Metro Nature’ as described by Wolf and Robbins [7]. Nature-based solutions (NBS), which include natural materials and processes, can also provide NEs in urban areas [8]. Following the modern taxonomy of NBS [8], they can be broadly categorised through their various uses: for stormwater management, soil and water remediation and bioengineering, greening systems to improve biodiversity in built environments, food and biomass production systems, and the provision of green space for human use. 

In addition to intrinsic values, NEs have a number of extrinsic values, providing landscapes, flora and fauna that supply ecosystem services [9], resulting in breathable air, water, and food production. NEs also provide recreational spaces, cultural capital, and serve as a genetic and biochemical repository for novel materials and pharmaceutics [10,11]. Human redevelopment of natural environments is a predominant cause of habitat disruption and is expected to reduce current biodiversity by 20–30% within 100 years [12]. 

Interactions with NEs provide positive cognitive, emotional and physical health outcomes [7,11,12,13,14,15]. Health benefits are associated with both ‘organic’ NEs that pre-date human influence and extend to ‘built’ NEs within urban areas, including parks, gardens and man-made wetlands [7,13,14,15,16,17,18]. Human exposure to NEs can take the form of passive interactions such as viewing greenspace through building windows, or active interactions where there is an immersive physical presence [7,13,14,15,16,18,19]. 

Living in urbanised areas with scarce NE elements is increasingly common; by 2040, 85% of the population in developed countries is predicted to be urban [12,20]. Urban living may contribute to immunoregulatory dysfunction by reducing human exposure to biodiversity, microbial diversity especially [21,22,23,24,25], furthering the rise of inflammatory mediated chronic health issues [11,12]. Autoimmune diseases, allergies, cancers, obesity, and type 2 diabetes may be associated with inflammatory conditions [11,12,21,23,26], which may, in turn, impose burdens on individuals and healthcare systems [12,27]. For example, long term elevation of inflammatory markers such as C-reactive protein (CRP) is associated with an increased risk of cardiovascular complications and mental health issues such as depression [11,21]. The links between improved mental health and exposure to NEs are well established [28,29] though the mechanistic connections are not always clear.

Given the health benefits of nature exposure, programs that foster nature engagement have potential co-benefits for health. Citizen Science projects often span environmental science fields, empowering participants to monitor flora and fauna in an effort to map trends, gain conservation literacy, and aid conservation efforts [30,31,32,33,34]. 

The identification of environmental changes often requires large-scale observations, which make citizen scientists a valuable resource [30,31,35]. As an example, participants generate upwards of ten million bird count observations annually for the Cornell Lab of Ornithology in New York [35]. New technologies allow scientists to utilise streamlined data management tools including websites and mobile applications which are customisable to specific research designs and allow integration of input from multiple collaborators, with simplified interfaces suitable for large scale data collection of Citizen Science [30,32,35,36]. Effective Citizen Science programs have high levels of support and guidance for citizen participants, including reference materials, and multiple lines of communication to research scientists, where open discussions are encouraged [30,34,35,36,37].

Citizen Science is a highly collaborative, multidisciplinary endeavour which is designed to achieve mutual benefits for participants and researchers [35,36,37,38,39]. It provides a platform for education, building scientific literacy through engagement with local environments [30,31,34]. Citizen Science initiatives that increase scientific literacy, generate meaningful connections to local environments, and further understanding of community views can also contribute to improved environmental policy making [31,36,37]. Integrating Citizen Science projects into the school curriculum has great student and teacher learning benefits, whilst encouraging interaction and connection with NEs [31,39]. Alliances between professionals in education, environmental sciences, statistics, and social sciences allow for both scientific and educational aims to be met [35]. Involvement in Citizen Science programs sparks discussions about environmental research in social interactions, reinforcing participant learning and broadening scientific literacy within the community [30,33,34].

Given the health impacts caused by urbanisation and the known health benefits of exposure to NEs, the development of initiatives to increase NE exposure in humans should be of public health benefit. Citizen Science is an increasingly used activity that may bring people into contact with NEs. In this review, we explore the current connections between NEs and human health and then interpret the potential value of Citizen Science as a mechanism for enhanced wellbeing. We then apply these findings to our argument for nature-based Citizen Science to be central to public health policy to enable systematic up-scaling of human exposure to NEs.

## 2. Methods

In order to develop a thesis that links urbanisation, natural environments and health, we conducted a narrative review. This focussed on two distinct sections: (a) urbanisation impacts on human health, and (b) NE interactions and human health. We then developed a schematic concept map that links these and illustrates the ways in which Citizen Science can foster NE preservation, and enhance human contact with NEs, and improve social interaction. Having established this, we then provide a discussion in which we propose the use of nature-based Citizen Science as a public health intervention in itself, which is ideally suited to those living in urban areas.

## 3. Results

### 3.1. Urbanisation and Human Health

Modifiable environmental risk factors are estimated to have accounted for 12.6 million deaths globally in 2010 and 22% of the worldwide disease burden in disability-adjusted life years [40]. Urbanisation has normalised sedentary routines and increased environmental pollutants, contributing to a rise in chronic and infectious diseases [10,11,15,18,26,40,41,42,43]. Built environments can contaminate waterways, and create excess noise that can be linked with adverse health outcomes beyond auditory effects [10,11,14,18,40], and generate air pollution which has robust associations with allergies, cancers, and mental health disorders [10,15,40]. Urban landscapes also generate a heat island effect attributable to a lack of vegetation and surface water, in conjunction with altered wind patterns, impermeable surfaces, heat generated from human activities, and air pollution which alters cloud cover [11,15,18,40,41,42,44,45,46]. Heatwaves are deadly natural disasters; for example, more than 70,000 people excess deaths were recorded in Europe due to the heatwave of 2003 [47]. Global surface temperatures are predicted to rise in the coming century [48] potentially amplifying existing health emergencies such as heat-related illnesses [10,41].

Mental health disorders affect one in five people globally each year [49], and are the biggest cause of disability worldwide [11]. Mental health disorders such as mood, anxiety and schizophrenic disorders have been associated with living in built, non-natural environments [1]. The psychological health of ageing populations in some countries (e.g., Australia, Sweden) is also an imminent issue, with an ever-growing risk of cognitive decline, increasing the strain on social support within communities along with the burden on the healthcare system [50,51]. Urban environments also generally contain fewer opportunities for people to engage with NEs which can impact health in many ways. The development of nature-based solutions for treating and remediating water and soils in cities [8] is acting to provide the range of Ecosystem Services [7] that facilitate exposure to NEs. 

### 3.2. Interactions between NEs and Human Health

Humans may respond in physiological, psychological and behavioural ways to NE exposure. The characteristics of NEs that may facilitate health benefits are likely to be multi-faceted, directly related to exposure to increased sunlight, air quality, biodiversity [including microbial], and phytochemicals in NEs, and indirectly through the opportunities NEs provide for restoration, socialisation, and physical activity.

Research demonstrates that exposure to NEs can enhance human cognitive performance in multiple ways; they can improve success within school and workplace settings, and have been used as a therapy tool to promote physical and emotional healing [7,14,16,17,18,19,29,40,52]. According to attention restoration theory (Table 1), NEs are restorative through redirecting attention, specifically the use of a passive soft fascination, contrasting the conscious attention required to meet the demands of busy urban living, triggering a physiological response that reduces stress and anxiety [13,14,16,17,29,53]. Stress Reduction Theory (Table 1) proposes ancestral preferences for NE properties associated with safety and resources remain relevant to unconscious stress-related neural mechanisms [19,29,54,55]. As little as 15 min of NE exposure is associated with lowered stress responses that are measurable from blood pressure, cortisol levels and pulse rate, with the most stressed individuals experiencing the greater stress-relief effect [29]. Passive exposure to NEs also produces restorative effects, generating greater capacities to concentrate attention in urban settings when living spaces have natural views, compared to ones overlooking urban artificial landscapes [29].

Greater accessibility to NEs is acknowledged as a key factor in promoting physical activity [7,11,13,14,17,18,46,56], however, it is insufficient on its own to encourage such behavioural changes [46,52]. Physical activity has important healthcare implications, with inactivity being the fourth highest contributor to worldwide mortality [52]. Physical activity provides an increased capacity to navigate stress as well as improved overall mental wellbeing [7,11,46,56] and is a highly encouraged preventative health measure against cardiovascular diseases [26,56,57]. Physical activity has also been shown to increase regulatory T cell activity, thereby limiting the inflammation associated with cardiovascular complications and mental health issues [21,26]. It has demonstrated a protective role against cognitive decline in later life, with findings suggesting a need for as little as three weekly walks to elicit this effect [57]. A study of an adult cohort aged over 65 years over a 5-year period demonstrated an ability of physical activity, at a frequency of three times a week, to improve learning and memory functioning by 42.3%, in addition to a 34% reduced risk of developing dementia [51]. Older adults have also demonstrated improvements in sit-to-stand and fast pace walking due to involvement in low-level volunteering [58], benefits that could also arise from Citizen Science activities. Whilst these benefits are associated with any form of physical activity, evidence suggests a rise in positive psychological reactions when undertaken within NEs [17,18,52].

Microbial flora (Table 1) such as bacteria, fungi, and protozoans coevolved with humans; exposure to diverse microbiomes builds our immune memory and educates and modulates our immune response [12,21]. As the diversity of microbial flora is diminished in urban environments [59,60] the protection they once provided from allergy, bowel inflammation and autoimmune diseases is also reduced [11,12,17,21,23,42]. The link between exposure to less diverse microbial communities and human health is described by two related hypotheses: the biodiversity hypothesis and the ‘old friends’ hypothesis (Table 1). The human gut alone contains over 160 species of bacteria, with both commonality and variation in species across individuals [12,61]. Human skin can have commensal relationships with many types of microorganisms including bacteria, fungi, viruses and microscopic protozoans [12]. Diversity within commensal microbiota is hypothesised as a key protection mechanism against adverse inflammatory responses, with different microorganisms exciting varying levels of regulatory stimulation [22,23]. Studies of rural living have linked agricultural land to increased diversity of the microbiota on skin and surfaces, inversely associated with the prevalence of allergies and asthma [17,21,22,42,62]. 

A lack of microbial stimuli disrupts immunoregulatory actions as evidenced by reductions in the regulatory cell activity of dendritic cells, T cells, and cytokines such as IL-10 and transforming growth factor-beta [12]. Gammaproteobacteria have been identified for immunoregulatory properties, being positively associated with lower allergy risk when commensal microbiota diversity is improved, thought to be a product of increased stimulation of anti-inflammatory IL-10 cytokines secreted from peripheral blood mononuclear cells [22]. Reduced organism biodiversity within surface and mattress dust has been associated with a greater risk of asthma in several studies of childhood environments [21]. The addition of a dog within the household from an early age has been found to reduce immunoglobulin E (IgE) sensitisation, protecting against allergy, likely a function of an increased microbiota diversity of household dust [11,21].

NEs that support social interactions are commonplace worldwide [13,16,38], meaning they could become a plentiful resource as a therapeutic treatment (aka ‘social prescription’), thus reducing the economic burden at the individual and community level [16]. This is one of the arguments for social prescribing [where medical professionals prescribe social services or activities rather than medical interventions] generally and ‘green’/nature prescribing specifically. Citizen Science programs could fall into this category as they can be socially engaging, with the potential to alleviate anxiety and promote better mental health, in addition to improving cognitive outcomes particularly among older participants [36,51,63]. Isolation from NEs and a reduction in diverse human contact associated with older age and reduced mobility is associated with elevated inflammatory markers including IL-6, and a dramatically reduced diversity of gut microbiota [11,21]; a finding of great concern in an ageing population [40,58]. Additionally, a limited social network is associated with a 60% increased risk of developing dementia within retired populations [51]. Citizen Science provides an opportunity for the mutual benefits of shared knowledge across generations, with the young learning from the experience of older participants, and reciprocating with an ability to help older citizens to improve their technological literacy and engagement [51].

### 3.3. Citizen Science, Natural Environments, and Urbanisation: Linkages Influencing Human Health 

The complexity and multi-faceted nature of NE exposure and human health, in the context of urbanisation, is schematically depicted (Figure 1). In this schematic, the negative consequences of urbanisation (depicted in red) are connected with NEs and human health. Urbanisation alters NEs, reduces biodiversity, and impacts human health. Conversely, exposure to NEs may improve human health. Citizen Science is shown as an agent to improve NEs through the generation of environmental research, gains in scientific literacy, and indirectly through policy change that acts to generate new NEs (such as through nature-based solutions in urban areas), or through preservation of existing areas. In addition, Citizen Science also promotes contact with NEs, which in turn may benefit human health (Figure 1). These benefits could be manifested through improved social connectivity, exposure to NEs directly impacting immune function, and increased exercise. Quantifying such benefits of NE exposure via Citizen Science has not been the subject of research thus far. By illustrating these links between NEs and human health, we propose future research (such as through clinical trials) that specifically links health gains with nature-based Citizen Science. 

## 4. Discussion: Nature-Based Citizen Science as Public Health Policy: Enticing Urban Dwellers into NEs

Many benefits of NEs are subconscious [7,68], which may present as a barrier when encouraging the community to connect with NEs. Nature-based Citizen Science projects can provide motivation for people to engage with NEs and have co-benefits for health through social interactions, physical activity and exposure to greater biodiversity. Unfamiliarity or past negative experiences with NEs can elicit negative associations, such as anxiety, uncertainty, or fear of aggressive or poisonous wildlife [11,18,29]. Evidence indicates that intentions to engage in certain behaviours, such as physical activity, can be improved following positive experiences of those behaviours [52,68]. Citizen Science projects may provide a catalyst for initial contact with NEs to shape participant attitudes for enduring attachment and engagement with nature [30].

The perceived social normality of interacting with NEs also has a critical influence on the attitudes and behaviours of individuals. Implementation of Citizen Science programs may require simultaneous public marketing and education strategies that help shift and shape perceived norms regarding interaction with NEs. For instance, the health benefits of Citizen Science are not routinely publicised in participant recruitment for projects; yet, such publicity may in fact increase participation and subsequently perceived norms around NE engagement. The development of empirical evidence linking Citizen Science and participant health may ultimately inform public health policy and Citizen Science practice itself, and introduce a sense of social encouragement to engage with NEs [52].

Despite established links between human health and NEs, several variables of therapeutic potential remain unclear, such as required exposures (akin to ‘dosages’), the duration of benefit after daily activities resume, and the impact of repeated exposure as familiarity with NEs are increased. The nature of interactions with NEs, and the influence they may have on protective health outcomes is also unclear. Finally, the impact of NE type (e.g., garden vs. park vs. forest) on therapeutic outcomes is unclear as is the wellbeing contribution of the NE exposure per se compared to the removal of negative urban stimuli. Answers for these questions would help inform more strategic and effective use of social/green/nature prescribing overall and the use of Citizen Science for health.

Nature-based Citizen Science provides a mechanism by which people may be exposed to NEs through systematic, organised and scalable activity. These activities provide multiple benefits and may be used to achieve a variety of scientific, conservation and educational goals. To the best of our knowledge, Citizen Science does not form an explicit component of health strategic planning.

Some jurisdictions, however, do have Citizen Science strategic plans and legislation, but not specifically for public health reasons. In Europe, there are a few national-level strategies emerging from a heterogeneous ecosystem of Citizen Science projects [69]. Links between Citizen Science and policy development are championed by COST (European Cooperation in Science and Technology, Available online: https://www.cost.eu/ (accessed on 7 December 2021)). In the USA, the American Innovation and Competitiveness Act (2017) contained provisions to utilise nationally coordinated Citizen Science to enhance scientific research, literacy and diplomacy. In Australia, the Inspiring Australia initiative (Available online: https://www.industry.gov.au/funding-and-incentives/inspiring-australia-science-engagement-in-australia (accessed on 7 October 2021)) aims to engage people with science, and as a consequence Citizen Science activities are enabled through this program. Amongst a diverse and highly contextualized body of Citizen Science projects, human health is often mentioned, but only in relation to particular projects that collect environmental data to improve health, not in the sense of Citizen Science activities being intrinsically healthy to do. 

Therefore, whilst Citizen Science is becoming part of government policy and strategy, this seems to contribute principal evidence to enhance environmental and public health objectives. Citizen Science is not yet an explicit part of public health policy, and we contend that embedding Citizen Science into public health strategy, particularly projects that facilitate NE engagement, could result in diverse health improvements for participants (e.g., physical, social, cognitive, etc.), while advancing science engagement and our scientific knowledge of the Australian environment. 

Linking nature-based Citizen Science to public health strategy could involve nominated targets for community involvement, set out in aspirational targets. Governments at both the state/provincial and national level are fond of establishing strategies that nominate key actions to improve health and wellbeing. We contend that this could include the explicit naming of nature-based Citizen Science as part of a strategy, with concomitant funding and directed recruitment of participants. 

However, before any further public investment in nature-based Citizen Science, demonstrating the explicit health benefits obtained from such activities is a vital next step. If nature-based Citizen Science can be shown to have health benefits, and these can be quantified in terms that can be linked to government aspirational strategy, then recommendations can be made to health authorities for systematic investment and incorporation with policy. We contend that clinical trial research should demonstrate health improvements in the domains such as overall quality of life and social connection, physical activity and over-the-counter (non-prescribed) medication use. Indeed, to this end, there are trials currently being conducted in Australia examining the health benefits of nature-based Citizen Science (e.g., the authors conducting a study for the South Australia Office for Ageing Well). 

## 5. Conclusions

The human health benefits of exposure to NEs are well established and can be achieved through participation in nature-based Citizen Science. This route of exposure to NEs could promote health through all three of the main pathways connecting health and nature: through increased (a) physical activity, (b) social interactions and (c) exposure to an increased quantity and diversity of microbiota. Exposure to diverse microbiomes is shown to aid in the development and maintenance of immunoregulation and changes to microbiomes and immune function can occur rapidly from exposure to NE. Long-term monitoring of C-reactive protein can determine the influence of NE exposure on inflammation over time, which can indicate levels of risk for cardiovascular diseases, inflammatory disorders, depression, and stress resilience and could be used to monitor the effectiveness of Citizen Science programs in reducing chronic stress and related conditions. 

Nature-based Citizen Science projects have the potential to motivate communities to further engage with NEs, providing holistic benefits to human health and the healthcare system, whilst generating the scientific research needed to better sustain NEs. Incorporating Citizen Science into public health policy will make the links between Citizen Science participation and health more explicit, thereby encouraging Citizen Science uptake and creating benefits to both public health and science.

## Figures and Tables

**Figure 1 ijerph-19-00068-f001:**
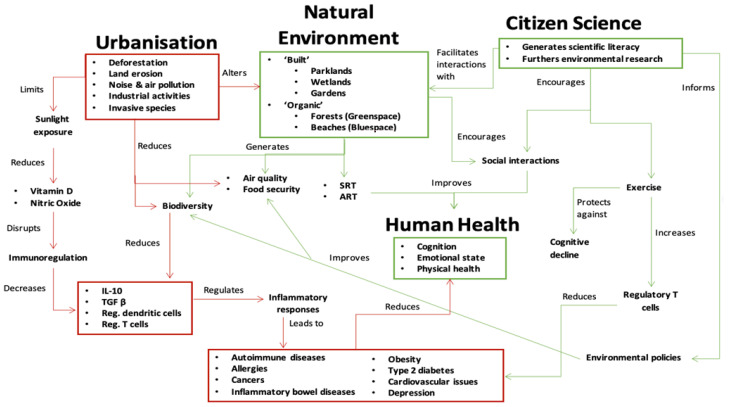
The complex relationships between Citizen Science, environmental exposure and human health. Evidenced and putative links are represented with arrows coloured in green (benefits), and red (harms). Abbreviations: ART: attention restoration theory IL-10: interleukin 10; SRT: stress reduction theory; TGF-β: transforming growth factor-beta.

**Table 1 ijerph-19-00068-t001:** Theories linking natural environments and human health, with details of key supportive experimental trials.

Theory	Participants	Study Design	Evidence	Reference
Kaplan’s attention restoration theory [ART] Urbanised living demands high levels of directed attention, taxing neural pathways that maintain focus and attention, NEs are hypothesised to stimulate soft fascination, switching unconscious neural processes and restoring tired pathways [29,53,64]. Core concepts theorised to influence the attention restoration mechanism are: a sense of being away, extent, fascination, and compatibility [29,53,64].	N = 12 8 males, 4 females Mean age = 30 Edinburgh University students.	A walk through three distinct districts, representing urban living [zone 1], a natural environment [zone 2], and a busy commercial zone [zone 3]. A neural cap recorded Electroencephalgram (EEG) data.	The zone 2 to zone 3 transition had decreased levels of arousal, frustration and engagement. Indicating that the natural environment reduced directed attention.	[53]
	N = 110 Predominantly staff and students of Chung-Hsing University.	Viewing 4 sets of 3 images, representative of the core concepts of ART; being away, extent, fascination, and compatibility. Images were viewed in 10s intervals, with 10s of non-viewing, blue screen in between [64]. Electomyography (EMG), EEG and blood volume pulse (BVP) measurements were taken.	Statistically significant EEG elevations, and decreased BVP, occurred whilst observing natural elements. Supporting claims that humans generate a response to elements within NEs.	[64]
	N = 38 23 males, 15 females Age = 18+ All living, working or studying in an urban, west midlands region of the UK [65].	30 min walks along 3 different trails; quiet residential streets [urban], inner-city parklands [greenspace], and along a canal [bluespace]. Measurements were taken at baseline [T1], after walking [T2], and 30 min later [T3]. Measurements included participant rated scales, backward digit spans, cortisol levels from saliva sampling, and heart rate monitoring.	The green and blue NEs gave greater cognitive function improvements and restorative experiences. Improvements in cognitive function took time to exhibit, being measurable at T3 but not T2.	[65]
	N = 12 7 females, 5 males Age = 18–24 undergraduate students of McMaster university.	Participants took photos of elements, within a natural place of their choosing, which they believe to positively contribute to their mental health. In depth interviews were used to collect data.	All participants expressed a correlation between removing themselves from built environments and improvements in mental health.	[13]
Ulrich’s stress reduction theory [SRT] The notion that elements of NEs can unconsciously trigger physiological and psychological stress reduction mechanisms, thought to be a remnant of survival instincts towards geographical preferences during human evolution [29,55].	N = 158 80 males, 78 females Age = 18–32 Long term US residents [55].	Self-reported stress was measured via Visual Analogue Scale. Stress was triggered with a Trier Social Stress Test. A personal viewing headset displayed one of ten 6-min 3D videos of street scenes with varying tree density.	Videos with higher tree density correlated with an increase in stress reduction. Tree cover at 62% density increased stress recovery by 60%, compared to a 2% density.	[55]
	N = 48 Young males.	15 min sitting in an urban and a forest landscape. Ongoing physiological measurements were taken as well as psychological self-reposts.	Forest areas significantly lowered diastolic blood pressure and heart rate and increased parasympathetic activity.	[54]
‘Old Friends’/biodiversity hypothesis A reduction in immunoregulation from a limited exposure to the microorganisms humans coevolved with, depriving the immune system of the input needed for education [11,12,17,21,22,23].	N = 60 Age = 7–14 50% living on traditional Amish farms, 50% living on industrialized Hutterite farms	Blood samples were collected from children along with history of allergies and asthma. Dust samples were collected from childrens’ bedrooms. Mice were exposed to the dust; immune and airway responses were monitored.	Amish children had 4–6 times lower prevalence of asthma and allergies and different innate immune cell composition. Amish dust had 6.8 times higher levels of endotoxin. Mice exposed to Amish dust had inhibited airway hyperreactivity; this protective effect was blocked in mice deficient in certain innate immune signals [MyD88 and Trif].	[66]
	N = 24 Healthy Canadian full term infants [61].	Gene sequencing, from stool samples taken at 3 months old, indicated microbiota composition. Mothers reported on the presence of siblings and household pets.	Microbiota quantity and diversity was increased for infants living with pets but not siblings. Siblings and pets altered the composition of the microbiota.	[61]
	GABRIELA N = 444 Age = 6–12 16% living on farms Rural Germany [62]	Settled dust in children’s bedrooms collected for culturing, gram staining and microscopy. Lung function testing with spirometry.	Samples from farming households had a higher biodiversity of fungi and bacteria, which correlated to a reduced prevalence of asthma.	[62]
	PARSIFAL N = 489 Age = 6–13 52% living on farms Rural Germany [62]	Mattress dust collected for single-strand conformation polymorphism testing.		
	Human blood samples and live mice	Dust was collected from an urban house and a farm barn and the microbial diversity was quantified. Monocyte-derived human dendritic cells [moDCs] were exposed to dust then coculture with purified naïve T cells. Mice were exposed to dust via intranasal administration.	Urban house dust contained a lower diversity of bacteria than farm barn dust. Exposure to urban house dust drove moDCs towards an ‘allergic’ [Th1-dominated response] while exposure to the highly diverse barn dust drove these cells towards a Th2-type response. Mice exposed to urban house, but not farm barn dust developed allergic inflammation in lungs.	[67]
	Genetically similar piglets [23].	Grown in isolation. Environmental exposures; sterile indoor environment, with or without antibiotics, and outdoors.	Microbiota compositions were dramatically impacted by alterations of early life environmental exposures [23]. Indoor grown piglets displayed upregulation of MHC-class 1 and various chemokines. Many of the identified phylotypes from this study can be found in humans.	[23]

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
