# Peer review of "Nature-Based Citizen Science as a Mechanism to Improve Human Health in Urban Areas"

_ijerph, 2021, doi:10.3390/ijerph19010068_

Round 1

Reviewer 1 Report

This is an interesting work that presents how contact with nature can improve mental and physical health and well-being by reviewing predominant theories on the subject. It also recommends integrating nature-based science into public health policies to improve the mental and physical well-being of urban citizens and government public health policies and strategies. I believe this could be an interesting and useful work for scholars and practitioners in urban planning, architecture, public health, geography, and design after some editing. Some improvements in terms of the methodological approach and analytical section, as well as a better link between evidence and results, would be needed to reach the full potential of the work and its contribution to the exisitng knowldge, as, in my opinion,  the article in its present form does not make a substantial contribution to either nature-based science research or methodology.

Tightening the theoretical framing of the paper and providing more in-depth methodology and analysis would enhance and clarify the contribution. The analysis provides rather less thorough insights into the understanding and the significance of nature-based science research than a traditional literature review does. A bibliometric analysis of specific cases would be a good underpinning for the literature review and the methodolical part of this paper.

Author Response

We are grateful for the Reviewer's comments. Below is a point-by-point account of our responses.

Reviewer: 'This is an interesting work that presents how contact with nature can improve mental and physical health and well-being by reviewing predominant theories on the subject. It also recommends integrating nature-based science into public health policies to improve the mental and physical well-being of urban citizens and government public health policies and strategies. I believe this could be an interesting and useful work for scholars and practitioners in urban planning, architecture, public health, geography, and design after some editing. Some improvements in terms of the methodological approach and analytical section, as well as a better link between evidence and results, would be needed to reach the full potential of the work and its contribution to the exisitng knowldge, as, in my opinion,  the article in its present form does not make a substantial contribution to either nature-based science research or methodology.

Tightening the theoretical framing of the paper and providing more in-depth methodology and analysis would enhance and clarify the contribution. The analysis provides rather less thorough insights into the understanding and the significance of nature-based science research than a traditional literature review does. A bibliometric analysis of specific cases would be a good underpinning for the literature review and the methodolical part of this paper.'

Response: We thanks the Reviewer for their comments, and agree that we needed a better theoretical framing of the paper. The following changes have been made in response:

1. A link between methods and results. The paper has been restructured to include a Methods and Results section, which hopefully illustrates our approach and findings.

2. Additional text which links the concepts of Citizen Science, Urbanization and Natural Environments exposure has been added to the bottom of page 7.

Reviewer 2 Report

You made a generalization which is not applicable. Your paper is specific to the Australian region, and you should keep it at that. This can be found on line 22 in the abstract where you stated “…. NEs do not form part of public health policy”. Here, you should indicate the context.

You should also be more specific with your statements. For instance, on line 128 you mentioned “… in some countries is also an”. Here it will be better to mention the countries instead of leaving the reader guessing.

Also, on line 139 you indicated “It has been suggested that …..”Here, it will also be better to remove any doubts by saying that “Research has suggested that………….

Page numbers

Page numbers on some pages do not go with the rest of the paper. These can be found mostly in the sections that contain Table 1. Check it out and make changes appropriately.

Method of selecting papers for review

The studies selected for the review in this paper provide evidence of the impact of citizen science and NEs on human health. However, the authors did not specify how they collected the articles they analyzed in this paper. I think it is important to show how those papers were selected, the inclusion criteria and the methods of evaluating the quality of those papers. Without these understanding readers will be left contemplating whether the papers are hand picked to fulfil the research or objectives or they are objectivity sampled. Thus, there is the need for the researchers to be explicit about their methods for selecting the articles for reviews.

No connection among the concepts in the paper

The paper utilizes three concepts: Citizen Science, Urban areas, and NEs. However, there are no connections among these three. The authors should make explicit connection among these three items at least in the introduction and conclusion section to drive home the main idea of the paper. Readers will be looking for this connection in relation to the title. In its current state, it seems that there are no connections among the three most essential aspects of your paper.

Author Response

We are grateful for the Reviewer's comments. Below is a point-by-point account of our responses.

Reviewer comments, followed by responses.

You made a generalization which is not applicable. Your paper is specific to the Australian region, and you should keep it at that. This can be found on line 22 in the abstract where you stated “…. NEs do not form part of public health policy”. Here, you should indicate the context.

Response: the Reviewer is correct, and specific discussion of the Australian context has been altered to provide an international analysis.

You should also be more specific with your statements. For instance, on line 128 you mentioned “… in some countries is also an”. Here it will be better to mention the countries instead of leaving the reader guessing.

Response: specific exemplar countries are now provided (pg 3).

Also, on line 139 you indicated “It has been suggested that …..”Here, it will also be better to remove any doubts by saying that “Research has suggested that………….

Response: text amended as suggested. Thank you for this comment.

Page numbers

Page numbers on some pages do not go with the rest of the paper. These can be found mostly in the sections that contain Table 1. Check it out and make changes appropriately.

Response: now corrected. It was a problem with page numbers re-starting in new sections.

Method of selecting papers for review

The studies selected for the review in this paper provide evidence of the impact of citizen science and NEs on human health. However, the authors did not specify how they collected the articles they analyzed in this paper. I think it is important to show how those papers were selected, the inclusion criteria and the methods of evaluating the quality of those papers. Without these understanding readers will be left contemplating whether the papers are hand picked to fulfil the research or objectives or they are objectivity sampled. Thus, there is the need for the researchers to be explicit about their methods for selecting the articles for reviews.

Response: Thanks for this comment and for pointing out something we can make more explicit. This manuscript contains a narrative review, and have not sought to produce a systematic or scoping review. Thus, paper selection following the normal conventions for narrative reviews to enable us to develop our thesis. We have explicitly described this as a Narrative Review (page 3)

No connection among the concepts in the paper

The paper utilizes three concepts: Citizen Science, Urban areas, and NEs. However, there are no connections among these three. The authors should make explicit connection among these three items at least in the introduction and conclusion section to drive home the main idea of the paper. Readers will be looking for this connection in relation to the title. In its current state, it seems that there are no connections among the three most essential aspects of your paper.

Response: we are grateful for this comment, and have now added a new section of the MS that links the three concepts (page 5).

Reviewer 3 Report

This paper represents important, evidence-based connections between health and citsci activities. 

Author Response

Reviewer: 'This paper represents important, evidence-based connections between health and citsci activities.' 

Response: we thank the Reviewer for this comment.

Reviewer 4 Report

The paper presented for the review deals with a very interesting issue of implementation of Citizen Science into a public health policy. However, the present structure of the paper is unclear.  The title suggests it's not only a review of state of art on topics, such as urbanization and human health, interactions between NEs and human health, etc., but something more. However when you read Introduction, followed by paragraphs 2, 3, 4, it's just a review. It should be commended that it is done thoroughly and cross-sectionally, but since the paper totally lacks a methodology it's difficult understand what is the goal. Is the review a goal in itself, or the the goal of the paper is hidden in Paragraph 4 (lines 269-273 and further)? So the biggest problem with the paper is it's structure and missing parts, i.e. Materials and methods, Results, Discussion. Moreover, the information you provide in the review part is comprehensive, but at the same time for readers knowing the subject it's nothing new. I'd advice to clearly define the goal of the research/ paper and present your hypothesis, methods you used to achieve the goal (e.g. review of literature and projects). Additionally, in the paragraph 4 you introduce some examples from Australia, but you never mention why you've chosen these particular ones. It should be also explained in the methodology. The paper is promising as it deals with important issue how to involve CS in the broader health policies, but you need to better present the scientific framework (by explaining what is the rationale of the research / explaining the research procedure/ presenting your findings/  and discussing them in the context of other studies/ concluding with suggestions). Technical issues: You need to change the referencing style as requirred by the MDPI - lines 174, 256-257, 260-261, 266-267).   In general, I think the paper is promising, but since it totally lacks description of research objective, methodology and results' discussion I think it requires major revisions.

Author Response

We are grateful for the comments from this reviewer. Below are the Reviewer's comments and our point-by-point responses.

Reviewer: 'The paper presented for the review deals with a very interesting issue of implementation of Citizen Science into a public health policy. However, the present structure of the paper is unclear.  The title suggests it's not only a review of state of art on topics, such as urbanization and human health, interactions between NEs and human health, etc., but something more. However when you read Introduction, followed by paragraphs 2, 3, 4, it's just a review. It should be commended that it is done thoroughly and cross-sectionally, but since the paper totally lacks a methodology it's difficult understand what is the goal. Is the review a goal in itself, or the the goal of the paper is hidden in Paragraph 4 (lines 269-273 and further)? So the biggest problem with the paper is it's structure and missing parts, i.e. Materials and methods, Results, Discussion. Moreover, the information you provide in the review part is comprehensive, but at the same time for readers knowing the subject it's nothing new. I'd advice to clearly define the goal of the research/ paper and present your hypothesis, methods you used to achieve the goal (e.g. review of literature and projects). Additionally, in the paragraph 4 you introduce some examples from Australia, but you never mention why you've chosen these particular ones. It should be also explained in the methodology. The paper is promising as it deals with important issue how to involve CS in the broader health policies, but you need to better present the scientific framework (by explaining what is the rationale of the research / explaining the research procedure/ presenting your findings/  and discussing them in the context of other studies/ concluding with suggestions). Technical issues: You need to change the referencing style as requirred by the MDPI - lines 174, 256-257, 260-261, 266-267).   In general, I think the paper is promising, but since it totally lacks description of research objective, methodology and results' discussion I think it requires major revisions.'

Response: we thank the Reviewer for these very useful and accurate comments. We have made the following changes in response.

  1. We have altered the structure of the MS so that the aims, methods and results are clear. At its heart there is a narrative literature review, but we now go on to make more explicit findings from this review.
  2. Referencing style: this has been reviewed and I believe it should now be consistent.
  3. International perspective: we have developed a more international perspective and discussion in the MS. These changes are principally in the Discussion section.

Reviewer 5 Report

The present Review Manuscript on "Nature-based Citizen Science as a mechanism to improve human health in urban areas" aims to gather the existing knowledge on human health mechanisms and their relation with natural environments (NEs) to promote Citizen Science programs within the health policies and strategies.

As a general recommendation/suggestion, I would have liked to find the concept of nature-based solutions (NBS) on the revision, since in urban environments, it is one of the forms of what the authors call as green spaces and/or natural environments (NEs). Please, try to connect this term within the content of the manuscript (i.e., abstract and main text).

A recent manuscript* would be of help to assist in this regards, as many other NBSs cited in the present review (e.g., urban forest, community gardens, green roofs, ...) are already classified in this study as NBS in urban environments. (*) Langergraber et al., 2021. A Framework for Addressing Circularity Challenges in Cities with Nature-Based Solutions. Water 2021, 13(17), 2355; https://doi.org/10.3390/w13172355

As a review, I recommend a more in-depth study of current references in this regard. From my point of view, this suggestion would give a contemporary focus to the article, helping to classify the examples of NEs as NBSs, in a clearer way. It gives the impression that there is a more complete and current research on the health side in relation to the topic of NEs.

As a reviewer, I value very positively the approach that the authors have given on the concept of Citizen Science, and how it is indispensable for the improvement of current urban policies. Similarly, I recommend the authors to cite and discuss published references on NBSs and education.

The term "Citizen Science" is found within the text as: "Citizen Science", "Citizen science", "citizen science"; please unify them.

Section 2 on "Urbanisation and human health" has to be extended, prioritizing potential benefits of NBSs on human health and well-being in relation to cities and their impacts on them, and by adding more examples and existing European Union policies on this sense.

Lines 156-163: try to reduce the term "Physical activity" in this paragraph for an easy read; and, for example, enumerate its benefits on bullet points.

Table 1. "EMG, EEG and BVP measurements were taken [67]." please, explain the acronyms. Table 1. Stean et al., 2016 instead of "Stein et al 2016", according to other reference citations. Table 1. References' numbers needed on Reference's column.

Line 203: "4. Mental health and social prescription", Line 220: Same section number "4. Nature-based Citizen Science as public health policy: enticing urban dwellers into NEs"; please, update it.

In my opinion, Section 4 needs to be extended as, although mention "social prescription" on its title, there is no so much information on this regards within the section's text.

In the last section on "Nature-based Citizen Science as public health policy: enticing urban dwellers into NEs", a more general vision on Citizen Science and public health policies is expected, and not only specific to the case of Australia. It is recommended to broaden this perspective or limit the Introduction and objectives of the article, only to the Australian experience; as mentioned in the Conclusions.

As a review, an increased number of cases studies and references are expected.

I do not see appropriate, citing references or including new Figures in the Conclusions when it could have been done within the body of the main text. Its modification is recommended. As a recommendation, Figure 1 would be placed as Supporting material or cited within the main body of the manuscript. Acronyms used in Figure 1 should also be described, e.g., in Figure caption.

It is recommended to re-write the Conclusions and within them not to cite acronyms, or if they are cited, do it in the same way as in the main text. For example, NEs instead of NE.

Lines 299-300: "Long term monitoring of CRP", please explain the acronym.

Author Response

We are grateful for the comments from this reviewer. Below we present the Reviewer's comments and our point-by-point response to them.

The present Review Manuscript on "Nature-based Citizen Science as a mechanism to improve human health in urban areas" aims to gather the existing knowledge on human health mechanisms and their relation with natural environments (NEs) to promote Citizen Science programs within the health policies and strategies.

As a general recommendation/suggestion, I would have liked to find the concept of nature-based solutions (NBS) on the revision, since in urban environments, it is one of the forms of what the authors call as green spaces and/or natural environments (NEs). Please, try to connect this term within the content of the manuscript (i.e., abstract and main text).

A recent manuscript* would be of help to assist in this regards, as many other NBSs cited in the present review (e.g., urban forest, community gardens, green roofs, ...) are already classified in this study as NBS in urban environments. (*) Langergraber et al., 2021. A Framework for Addressing Circularity Challenges in Cities with Nature-Based Solutions. Water 2021, 13(17), 2355; https://doi.org/10.3390/w13172355

Response: we thank the reviewer for this helpful comment and have now included NBS in the Abstract and Introduction as suggested. The suggested citation has been included.

As a review, I recommend a more in-depth study of current references in this regard. From my point of view, this suggestion would give a contemporary focus to the article, helping to classify the examples of NEs as NBSs, in a clearer way. It gives the impression that there is a more complete and current research on the health side in relation to the topic of NEs.

As a reviewer, I value very positively the approach that the authors have given on the concept of Citizen Science, and how it is indispensable for the improvement of current urban policies. Similarly, I recommend the authors to cite and discuss published references on NBSs and education.

The term "Citizen Science" is found within the text as: "Citizen Science", "Citizen science", "citizen science"; please unify them.

Response: thank you for pointing out these inconsistencies. All have now been corrected.

Section 2 on "Urbanisation and human health" has to be extended, prioritizing potential benefits of NBSs on human health and well-being in relation to cities and their impacts on them, and by adding more examples and existing European Union policies on this sense.

Response: specific reference to Nature-Based Solutions has now been included in this section. Thank you for the suggestion.

Lines 156-163: try to reduce the term "Physical activity" in this paragraph for an easy read; and, for example, enumerate its benefits on bullet points.

Response: text amended as suggested for greater readability (page 4).

Table 1. "EMG, EEG and BVP measurements were taken [67]." please, explain the acronyms. Table 1. Stean et al., 2016 instead of "Stein et al 2016", according to other reference citations. Table 1. References' numbers needed on Reference's column.

Response: terms now explained in Table 1.

Line 203: "4. Mental health and social prescription", Line 220: Same section number "4. Nature-based Citizen Science as public health policy: enticing urban dwellers into NEs"; please, update it.

Response: corrected.

In my opinion, Section 4 needs to be extended as, although mention "social prescription" on its title, there is no so much information on this regards within the section's text.

Response: specific naming of social prescription has now been added (page 5)

In the last section on "Nature-based Citizen Science as public health policy: enticing urban dwellers into NEs", a more general vision on Citizen Science and public health policies is expected, and not only specific to the case of Australia. It is recommended to broaden this perspective or limit the Introduction and objectives of the article, only to the Australian experience; as mentioned in the Conclusions.

As a review, an increased number of cases studies and references are expected.

Response: we thank the Reviewer for their perspective. However, we believe the manuscript contains the right balance between review, synthesis and perspective, and with 71 total citations, this balance has been struck.

I do not see appropriate, citing references or including new Figures in the Conclusions when it could have been done within the body of the main text. Its modification is recommended. As a recommendation, Figure 1 would be placed as Supporting material or cited within the main body of the manuscript. Acronyms used in Figure 1 should also be described, e.g., in Figure caption.

Response: agreed. The figure has been moved to the Results.

It is recommended to re-write the Conclusions and within them not to cite acronyms, or if they are cited, do it in the same way as in the main text. For example, NEs instead of NE.

Lines 299-300: "Long term monitoring of CRP", please explain the acronym.

Reviewer 6 Report

Reviewer’s General Comments:

The manuscript presents a review of literature on using citizen science to promote community contact with nature and increase wellbeing. The idea of the work is novel and highly relevant, but prior to publication there are some major (and minor) comments that need to be addressed.  

Comments:

Line number

Comment

1.      LN 28

“Human populations have become increasingly urbanised” – not sure about this phrase. How can “populations” become urbanised. Maybe change it to Human habitats, or environments.

2.      LN 220

Change the title numbering to “5.”

3.      LN 238-240

I agree that there needs to be stronger empirical evidence for this. However, I feel like this work could have started to shed some light on this evidence. I think this manuscript should include a section where it reviews INTERNATIONAL citizen-science projects and point to indirect or direct evidence of citizens showing improvements in health/wellbeing as a result of being involved in these studies. Without this section, this manuscript is missing one large component, since it has a good review of possible health benefits of NEs, but completely skips on reviewing citizen-science, and then jumps into (more-or-less) discussion and recommendations of Section 5. So I would add a section between 4 and 5, just reviewing different CS projects and their reported benefits to participants (table or paragraph form).     

4.      Ln 250-272

Not sure why you only reviewed Australian sources. Since this is international publication, you need to provide wider review of policies and programs across the world (probably good to put it under additional subheading).

5.      Conclusion section

I am not sure why you present this Figure in the conclusions. It should be presented earlier (likely in section 5), as well as some of the “Discussion” from the conclusions. Conclusion section should be kept free of citation (especially re-citation) and conclude the remarks brought forward in the body of the manuscript. It should not however add new information or figures/tables.

6.      Figure 1

As mentioned in the previous comment, figure 1 could become the focus of a discussion about potential framework for promoting the value of Citizen Science initiative. Consider the possibility of adding a short discussion on using this Figure (with some modifications) to “score” health benefits of individual CS projects. E.g. depending on how many “factors” projects take into account (exposure to nature, social interaction, etc.), they can be “scored” and promoted as more/less beneficial to participants. This could be a very strong contribution of this manuscript to International Science, and will likely increase citation of this work.

7.      LN 310

Again, not sure why the focus is only Australia. It needs to be made more international, considering you are discussing global phenomena.

8.      LN 22

Once more review on citizen science is added, I would add one more sentence here on the finding of that review.

Author Response

We are grateful for the Reviewer's comments and have responded point-by-point as detailed below:

“Human populations have become increasingly urbanised” – not sure about this phrase. How can “populations” become urbanised. Maybe change it to Human habitats, or environments.

Response: Wording amended as suggested (page 1).

Change the title numbering to “5.”

Response: title numbering amended.

I agree that there needs to be stronger empirical evidence for this. However, I feel like this work could have started to shed some light on this evidence. I think this manuscript should include a section where it reviews INTERNATIONAL citizen-science projects and point to indirect or direct evidence of citizens showing improvements in health/wellbeing as a result of being involved in these studies. Without this section, this manuscript is missing one large component, since it has a good review of possible health benefits of NEs, but completely skips on reviewing citizen-science, and then jumps into (more-or-less) discussion and recommendations of Section 5. So I would add a section between 4 and 5, just reviewing different CS projects and their reported benefits to participants (table or paragraph form).     

Response: We thank the Reviewer for these insightful comments. A more international perspective has been introduced (page 12). The links between Citizen Science and Natural Environment exposure have now been made more explicit (page 5).

Not sure why you only reviewed Australian sources. Since this is international publication, you need to provide wider review of policies and programs across the world (probably good to put it under additional subheading).

Response: We agree completely. An international perspective has now been introduced (page 12).

I am not sure why you present this Figure in the conclusions. It should be presented earlier (likely in section 5), as well as some of the “Discussion” from the conclusions. Conclusion section should be kept free of citation (especially re-citation) and conclude the remarks brought forward in the body of the manuscript. It should not however add new information or figures/tables.

Response: The Figure has been moved to a more appropriate section.

As mentioned in the previous comment, figure 1 could become the focus of a discussion about potential framework for promoting the value of Citizen Science initiative. Consider the possibility of adding a short discussion on using this Figure (with some modifications) to “score” health benefits of individual CS projects. E.g. depending on how many “factors” projects take into account (exposure to nature, social interaction, etc.), they can be “scored” and promoted as more/less beneficial to participants. This could be a very strong contribution of this manuscript to International Science, and will likely increase citation of this work.

Response:

Again, not sure why the focus is only Australia. It needs to be made more international, considering you are discussing global phenomena.

Response: An international perspective has now been introduced (page 12).

Once more review on citizen science is added, I would add one more sentence here on the finding of that review.

Response: Our principal focus is on human exposure to natural environments, using citizen science as a lever. There isn’t much information on the health benefits of citizen science, hence the need for this review. Text has been added which explicitly calls out the need for future research to quantify nature-based citizen science and health benefits (page 5).

Round 2

Reviewer 4 Report

The revised version of the MS is much better. The new structure and clear description of aims and methods improved the paper a lot. My only advice is  to read the text through in order to eliminate repetitions (e.g. in abstract line 13, a double "include" - maybe you can rephrase it) and polish the style.

Author Response

The revised version of the MS is much better. The new structure and clear description of aims and methods improved the paper a lot. My only advice is  to read the text through in order to eliminate repetitions (e.g. in abstract line 13, a double "include" - maybe you can rephrase it) and polish the style.

Response: Thank you for your feedback. Text in line 13 has been altered, and other in-text changes made to remove repetitions.

Reviewer 5 Report

To the Authors:

Please, see below some comments regarding this Second Round of Review:

  • The authors, although confirming the modifications after the review, have not carried out some of the suggested recommendations.

For instance:

"Lines 156-163: try to reduce the term "Physical activity" in this paragraph for an easy read; and, for example, enumerate its benefits on bullet points."

"Response: text amended as suggested for greater readability (page 4)."

  • In this case, it can be accepted that no numbering or bullet points are used. However, the authors did not justify their action.

However, in Table 1:

"... Table 1. References' numbers needed on Reference's column."

  • The authors have not numbered the references in the last column of the Table 1, according to the numbering followed in the text and in the References section.

Following previous review' comments:

"In the last section on "Nature-based Citizen Science as public health policy: enticing urban dwellers into NEs", a more general vision on Citizen Science and public health policies is expected, and not only specific to the case of Australia. It is recommended to broaden this perspective or limit the Introduction and objectives of the article, only to the Australian experience; as mentioned in the Conclusions.

As a review, an increased number of cases studies and references are expected."

"Response: we thank the Reviewer for their perspective. However, we believe the manuscript contains the right balance between review, synthesis and perspective, and with 71 total citations, this balance has been struck."

  • The authors do not answer the first question about the specific case of Australia. Please give reasons why this recommendation / suggestion by the reviewer is not being followed.

"I do not see appropriate, citing references or including new Figures in the Conclusions when it could have been done within the body of the main text. Its modification is recommended. As a recommendation, Figure 1 would be placed as Supporting material or cited within the main body of the manuscript. Acronyms used in Figure 1 should also be described, e.g., in Figure caption."

"Response: agreed. The figure has been moved to the Results."

  • Authors did not follow last recommendation: "Acronyms used in Figure 1 should also be described, e.g., in Figure caption.".
  • The authors, in the case of not following the proposed recommendations, are asked to justify their decision.

Author Response

Responses to Reviewer 5 are as below:

To the Authors:

Please, see below some comments regarding this Second Round of Review:

  • The authors, although confirming the modifications after the review, have not carried out some of the suggested recommendations.

For instance:

"Lines 156-163: try to reduce the term "Physical activity" in this paragraph for an easy read; and, for example, enumerate its benefits on bullet points."

"Response: text amended as suggested for greater readability (page 4)."

  • In this case, it can be accepted that no numbering or bullet points are used. However, the authors did not justify their action.

RESPONSE TO ROUND 2 comments: Thank you for your feedback, whilst we acknowledge that bullet points and/or numbering can assist in readability, it can also impact the overall flow. We believe in the case, leaving the text as paragraphs allows for a more cohesive narrative to be told to the reader. 

However, in Table 1:

"... Table 1. References' numbers needed on Reference's column."

  • The authors have not numbered the references in the last column of the Table 1, according to the numbering followed in the text and in the References section.

RESPONSE TO ROUND 2 COMMENTS: This has been amended, all references now numbered according to in text citations and the referencing section.

Following previous review' comments:

"In the last section on "Nature-based Citizen Science as public health policy: enticing urban dwellers into NEs", a more general vision on Citizen Science and public health policies is expected, and not only specific to the case of Australia. It is recommended to broaden this perspective or limit the Introduction and objectives of the article, only to the Australian experience; as mentioned in the Conclusions.

As a review, an increased number of cases studies and references are expected."

"Response: we thank the Reviewer for their perspective. However, we believe the manuscript contains the right balance between review, synthesis and perspective, and with 71 total citations, this balance has been struck."

  • The authors do not answer the first question about the specific case of Australia. Please give reasons why this recommendation / suggestion by the reviewer is not being followed.

RESPONSE TO ROUND 2 COMMENTS: Apologies for the oversight in the response. We have indeed provided greater international perspective and taken the focus away from Australia. This was done in response to comments from other reviewers, and it was an oversight not to specifically respond to this reviewer as well. Amended text can be found on pg 12, in which we discuss a more international perspective.

"I do not see appropriate, citing references or including new Figures in the Conclusions when it could have been done within the body of the main text. Its modification is recommended. As a recommendation, Figure 1 would be placed as Supporting material or cited within the main body of the manuscript. Acronyms used in Figure 1 should also be described, e.g., in Figure caption."

"Response: agreed. The figure has been moved to the Results."

  • Authors did not follow last recommendation: "Acronyms used in Figure 1 should also be described, e.g., in Figure caption.".
  • The authors, in the case of not following the proposed recommendations, are asked to justify their decision.

RESPONSE TO ROUND 2 COMMENTS:

This has been amended with acronyms now appearing in a figure caption. Refer to line 275.

Reviewer 6 Report

Consider the possibility of adding a short discussion on using this Figure (with some modifications) to “score” health benefits of individual CS projects. E.g. depending on how many “factors” projects take into account (exposure to nature, social interaction, etc.), they can be “scored” and promoted as more/less beneficial to participants. This could be a very strong contribution of this manuscript to International Science, and will likely increase citation of this work.

Author Response

Consider the possibility of adding a short discussion on using this Figure (with some modifications) to “score” health benefits of individual CS projects. E.g. depending on how many “factors” projects take into account (exposure to nature, social interaction, etc.), they can be “scored” and promoted as more/less beneficial to participants. This could be a very strong contribution of this manuscript to International Science, and will likely increase citation of this work.

Response: Thank you for your feedback. Whilst agreed, the figure could assist in creating a “score” of the health benefits of an individual undergoing CS projects, we feel this is beyond the scope of this narrative review, which was conducted to link natural environments, urbanisation and health. Such discussion around a scoring system would benefit from a separate/additional publication.